# Coherent acoustic control of a single silicon vacancy spin in diamond

Smarak Maity[1], Linbo Shao[1], Stefan Bogdanović[1], Srujan Meesala[1], Young-Ik Sohn[1], Neil Sinclair[1,2], Benjamin Pingault[1], Michelle Chalupnik[1], Cleaven Chia[1], Lu Zheng[3], Keji Lai[3] & Marko Lončar[1]*

Phonons are considered to be universal quantum transducers due to their ability to couple to a wide variety of quantum systems. Among these systems, solid-state point defect spins are known for being long-lived optically accessible quantum memories. Recently, it has been shown that inversion-symmetric defects in diamond, such as the negatively charged silicon vacancy center (SiV), feature spin qubits that are highly susceptible to strain. Here, we leverage this strain response to achieve coherent and low-power acoustic control of a single SiV spin, and perform acoustically driven Ramsey interferometry of a single spin. Our results demonstrate an efficient method of spin control for these systems, offering a path towards strong spin-phonon coupling and phonon-mediated hybrid quantum systems.

[1] John A. Paulson School of Engineering and Applied Sciences, Harvard University, 29 Oxford Street, Cambridge, MA 02138, USA. [2] Division of Physics, Mathematics and Astronomy, and Alliance for Quantum Technologies (AQT), California Institute of Technology, 1200 E. California Boulevard, Pasadena, CA 91125, USA. [3] Department of Physics, University of Texas at Austin, 110 Inner Campus Drive, Austin, TX 78705, USA. *email: loncar@seas.harvard.edu

Phonons, or mechanical vibrations, offer unique advantages for the transfer of quantum information between solid-state quantum systems. They couple to a wide variety of individual qubits[1–4], and are envisioned as universal quantum transducers between different types of qubits[5–7]. Among these, several defect spins in diamond and silicon carbide are not only optically accessible, but also possess long coherence times allowing them to be used as long-lived quantum memories[8–10]. Although control of some of these defect spins with mechanical vibrations has been demonstrated[3,11–16], obtaining strong spin-phonon coupling remains a challenge due to their low strain susceptibility. Recent experiments have shown the potential of the negatively charged silicon vacancy (SiV) center in diamond to act as a spin qubit that is highly sensitive to strain due to the orbital degeneracy in its ground state[17,18], making it particularly suitable for coupling to phonons. Here, we utilize this high strain susceptibility to demonstrate coherent, low-power control and Ramsey interferometry of a single SiV spin using surface acoustic waves (Fig. 1a). Our results establish the electron-phonon coupling of the SiV spin qubit as a promising approach for transduction between single spins and phonons. This would enable the

generation of quantum nonlinearities for single phonons, and realization of phonon-mediated hybrid quantum systems with spins.

## Results

**Strain response of the SiV.** The SiV is a point defect in diamond that has attracted attention due to its excellent optical properties[19], long-lived electronic spin qubit[10], and ability to maintain these properties inside nanoscale devices[20,21]. It is formed by a silicon atom located centrally between two adjacent vacant sites in the diamond lattice (Fig. 1b), and hence possesses $D_{3d}$ symmetry[22]. Its electronic structure is composed of a ground state (GS) and an excited state (ES), each split into two, principally by spin-orbit coupling, resulting in four optical transitions. The brightest of these is the transition between the lower branches of the ES and GS, labelled C, which we utilize in this work. In the presence of a magnetic field, the degeneracy of the S = 1/2 electronic spin of the SiV is lifted and each energy level further splits into two (Fig. 1c). In particular, the lower ground state splits into the levels $|e_{g+} \downarrow\rangle$ and $|e_{g-} \uparrow\rangle$, which are used as a long-lived spin qubit[10]. The lifting of the spin degeneracy causes each optical transition to split into four, as shown in Fig. 1d. In the case of transition C, this gives rise to two bright spin-conserving transitions, that we label C2 and C3, and two dim spin-flipping transitions, labelled C1 and C4.

The strain response of the SiV electronic states depends on the orientation of the strain with respect to the symmetry axis of the SiV[18]. Strain tensor components belonging to the $A_{1g}$ representation of the $D_{3d}$ symmetry group induce a global shift of the energy levels, while the $E_g$ representation components mix the orbitals. In the presence of an off-axis magnetic field, the mixing of the spin degree of freedom in the spin-orbit eigenstates allows $E_g$ strain to introduce a non-zero overlap term for the $|e_{g+} \downarrow\rangle \leftrightarrow |e_{g-} \uparrow\rangle$ qubit transition. As a result, the qubit levels inherit the large strain susceptibility resulting from perturbations to the spatial charge distribution of orbitals ($\sim 1$ PHz/strain). The resulting strain susceptibility for the spin qubit levels can be $\sim 100$ THz/strain for qubit transition frequencies in the few GHz range. This is four orders of magnitude larger compared to other defect spins whose acoustic driving has been studied recently[3,11,15,16]. In these latter cases, the strain susceptibility arises from a perturbation to the much weaker spin-spin interaction within the same orbital or with a far-detuned excited state orbital[14,23]. The fundamentally different origin of spin-phonon coupling in the SiV enables efficient control of the SiV spin qubit with an oscillating strain field.

**Surface acoustic devices on diamond.** Surface acoustic waves (SAWs) are travelling mechanical vibrations confined to solid surfaces, and can be thought of as bound states whose band-structures lie below the continuum of bulk acoustic waves. They are particularly suitable for interfacing with near-surface atomic-scale defects in solids[12,13,15], due to their inherent confinement of acoustic energy to a depth of about one acoustic wavelength (3 $\mu$m in our case). Our SAW devices (Fig. 1a) are fabricated on single-crystal diamond with the top surface normal to the [100] crystal direction. We generate SiVs at a depth of 100 ± 18 nm by implantation of silicon ions. The area density of SiVs is sparse enough that SiVs can be individually addressed optically. Due to the lack of piezoelectricity in diamond, which is required for electrical transduction of SAWs, we deposit a thin layer of piezoelectric aluminum nitride (AlN) on top of the diamond. We fabricate aluminum interdigital transducers (IDTs) on top of the AlN. These are patterns of interlocking metal fingers, with each

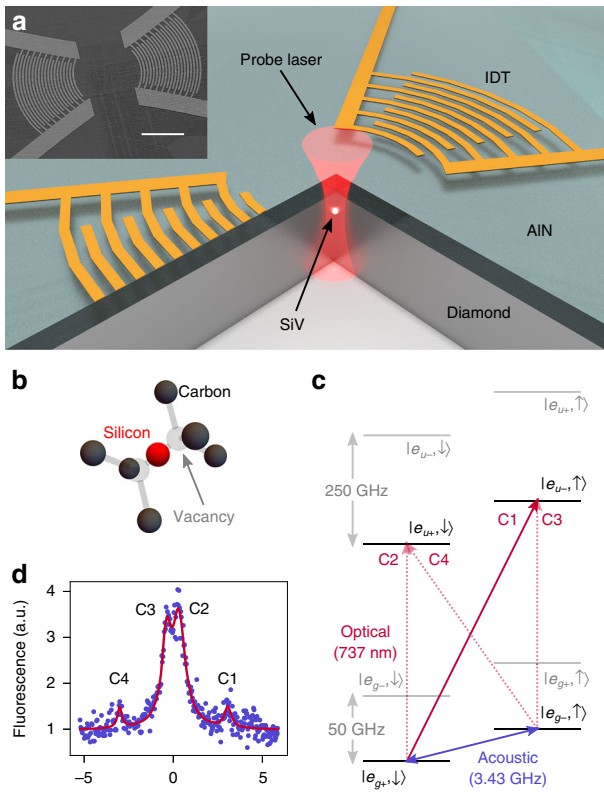

**Fig. 1 Layout of surface acoustic wave (SAW) devices, and structure of the silicon vacancy (SiV) center. a** Schematic of our diamond SAW device. A microwave signal applied to one of the interdigital transducers (IDTs) generates acoustic waves due to the piezoelectric response of aluminum nitride (AlN). SiVs in the diamond are probed using a focused laser beam. (Inset) A scanning electron microscope (SEM) image of a pair of transducers. The scale bar corresponds to 20 $\mu$m. **b** Molecular structure of the SiV. **c** Electronic structure of the SiV under non-zero external magnetic field. The solid red arrow indicates the optical transition used for spin initialization and readout, and the dashed red arrows indicate other optical transitions. The blue arrow indicates the acoustic transition between the two levels of the spin qubit. **d** Optical fluorescence spectrum of the C-transition of the SiV under resonant optical excitation, showing fine structure. The four peaks correspond to the transitions C1–C4.

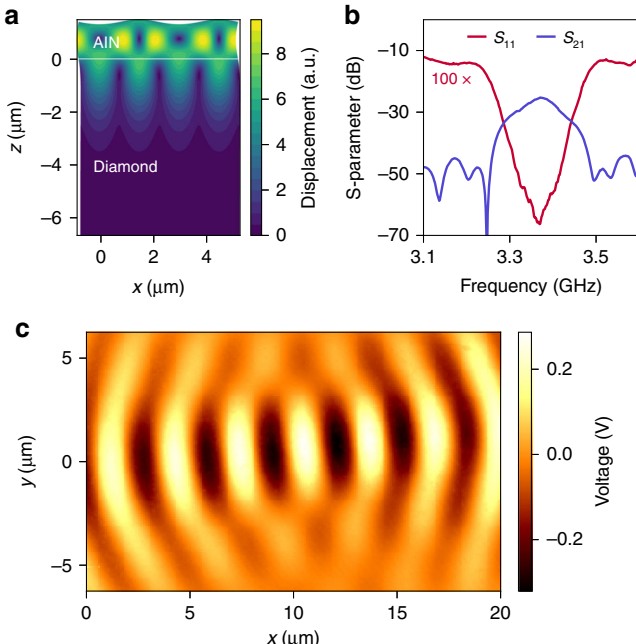

**Fig. 2 Characterization of SAW transducers. a** Total displacement profile of the acoustic mode in a cross-section of the device, obtained from finite element simulation. The white horizontal line indicates the interface between AlN and diamond. SiVs are located 100 nm below this line. **b** Room temperature measurement of electrical S-parameters between the transmitter and receiver IDTs. The $S_{11}$ plot is magnified 100× along the vertical axis. **c** Microwave impedance microscopic image showing the surface electric potential at the focus of the transducers under continuous-wave excitation. The potential is proportional to the SAW amplitude and demonstrates focusing of the SAW on the surface.

set of fingers connected to a common terminal. When an alternating voltage is applied to the terminals of the IDT, the piezoelectric response of the AlN generates spatially and temporally periodic deformations on the surface, which propagate as SAWs. We design the shape of our transducers to generate Gaussian SAW beams in order to concentrate acoustic energy by focusing. Fabricating transducers in pairs allows us to electrically measure their total scattering parameters (S-parameters).

We evaluate the acoustic mode profile of the SAWs with finite element simulations (Fig. 2a), confirming that the SAW is localized to a depth of one acoustic wavelength. Electrical S-parameter measurements indicate a center frequency of 3.37 GHz, with a full width half maximum (FWHM) amplitude bandwidth of 126 MHz that is limited by the number of finger pairs in the transducers (Fig. 2b). The shortest acoustic pulse generated by these devices is 13 ns in duration, as measured by time-domain characterization (Supplementary Note 2). We measure the surface amplitude profile of the SAWs using transmission-mode microwave impedance microscopy[24]. This technique uses a scanning probe to measure the surface electric potential on a piezoelectric material, which is proportional to the SAW amplitude at each point on the surface. The measured electric potential profile indicates that the SAW is focused to about one acoustic wavelength laterally (Fig. 2c), in addition to the localization in depth (Fig. 2a). In combination with the high strain susceptibility of the SiV, this acoustic confinement further reduces the acoustic power required to drive an individual SiV spin.

**Acoustic driving of the SiV spin.** After characterizing our devices at room temperature, we proceed with low-temperature

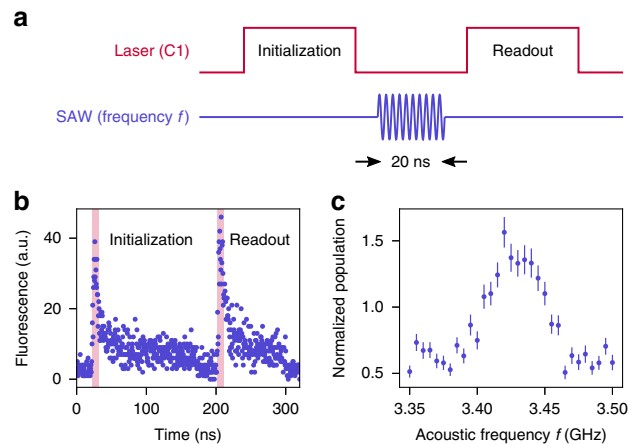

**Fig. 3 Optically detected acoustic resonance (ODAR) measurements of the SiV spin transition. a** Optical and acoustic pulse sequences used for ODAR measurement. The laser is resonant with the C1 transition and initializes the SiV into $|\uparrow\rangle$. A 20 ns duration SAW pulse drives population between $|\uparrow\rangle$ and $|\downarrow\rangle$. **b** Time-resolved histogram of photon detection events corresponding to the pulse sequence in **a**. The height of the peaks at the beginning of the initialization and readout pulses is proportional to the population in $|\downarrow\rangle$. Photon detections are integrated over 10 ns (indicated by the shaded region) to determine initialization and readout signals. **c** Normalized population in the $|\downarrow\rangle$ state as the center frequency of the acoustic pulse is varied, calculated as the ratio between the readout and initialization signals. A maximum is obtained at 3.43 GHz, indicating the resonance frequency of the SiV spin transition. The error bars represent the standard deviation of the normalized population.

acoustic driving of SiV spins. These experiments are performed at a temperature of 5.8 K, as measured on the cold stage inside a closed-cycle liquid helium cryostat. A confocal microscope focuses light from a tunable 737 nm laser on the sample, which is used to resonantly excite optical transitions of individual SiVs, with pulses generated using an electro-optic intensity modulator. We detect photons emitted in the phonon sideband (PSB) of the SiV emission spectrum. Our SiV spin qubit is defined by the levels $|e_{g+}\downarrow\rangle$ and $|e_{g-}\uparrow\rangle$ (Fig. 1c) (denoted $|\downarrow\rangle$ and $|\uparrow\rangle$ respectively for simplicity), and its transition frequency is adjusted to lie within the bandwidth of the acoustic tranducers (Fig. 2b) by tuning an external magnetic field.

To demonstrate acoustic driving of the SiV spin, we use optically detected acoustic resonance (ODAR). For our experiments, we select an SiV located near the focus of the transducers. An arbitrary waveform generator (AWG) drives one of the transducers to generate acoustic pulses. The acoustic pulses affect the SiV levels via both energy shift (dispersive) and mixing interactions with different components of the applied strain[18]. We utilize the dispersive interaction to calibrate the timing of the optical and acoustic pulses, taking into account the acoustic velocity (Supplementary Note 3). On the other hand, the mixing strain response resonantly drives population between the qubit states.

The optical and acoustic pulse sequences for ODAR are shown in Fig. 3a. We use optical pulses that are resonant with the spin-flipping C1 optical transition (Fig. 1c) for both spin initialization and readout. A 150 ns duration optical pulse initializes the SiV into $|\uparrow\rangle$, from the inital thermal mixture of $|\downarrow\rangle$ and $|\uparrow\rangle$. The corresponding time-resolved histogram of PSB photon detection events shows an initial peak proportional to the thermal population in the $|\downarrow\rangle$ state, followed by an exponential decay as population is transferred to $|\uparrow\rangle$ via the spin-conserving C3 transition (Fig. 3b). After initialization, a 20 ns duration acoustic

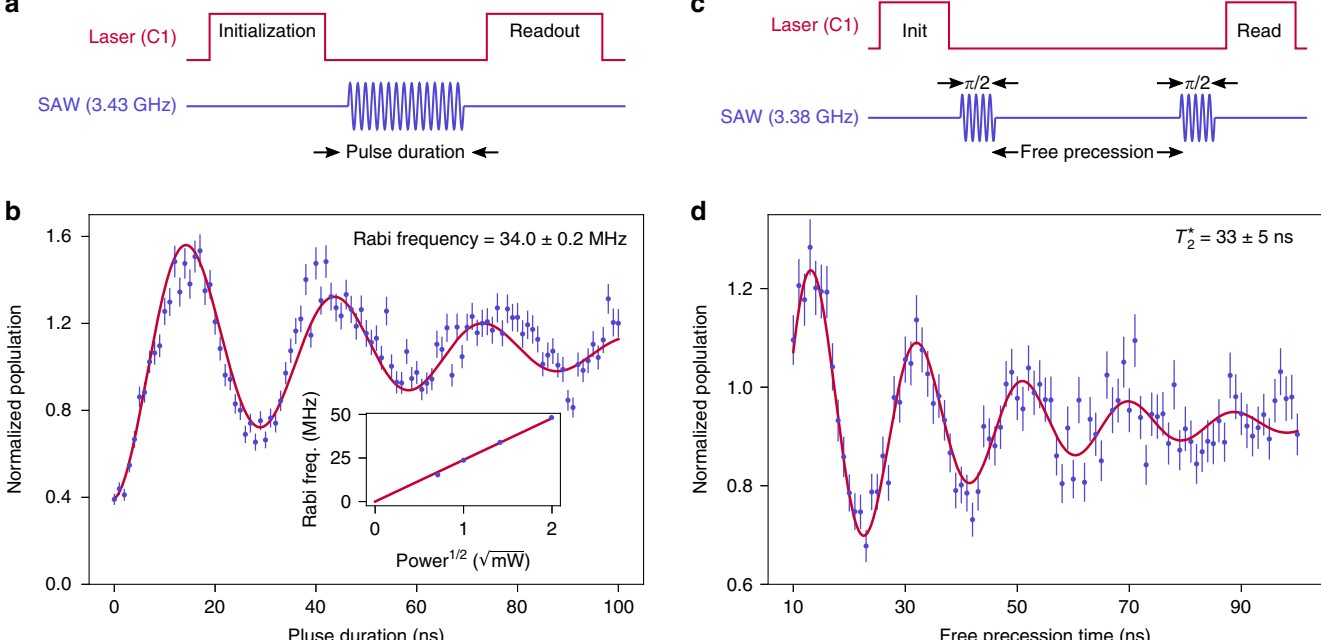

**Fig. 4 Coherent acoustic control of the SiV spin qubit. a** Pulse sequence used for Rabi oscillation measurements. An acoustic pulse of frequency 3.43 GHz, which is resonant with the SiV spin transition and generated with 2 mW peak microwave input power, is used to coherently drive the qubit. **b** Normalized population in the $|\downarrow\rangle$ state as the acoustic pulse duration is varied. A fit to a theoretical model is shown in red. (Inset) The dependence of Rabi frequency on the square root of peak microwave input power indicates the expected linear behavior. The errors in the Rabi frequencies are of the order of 0.1 MHz. **c** Pulse sequence used for Ramsey interferometry measurements. Two $\pi/2$ acoustic pulses, each detuned by 50 MHz and generated with 4 mW peak input power, are separated by a varying free precession time. **d** Normalized population in the $|\downarrow\rangle$ state as the free precession time is varied. A fit to a theoretical model is shown in red. The time constant of the exponential decay of the oscillations gives a spin coherence time $T_2^* = 33 \pm 5$ ns. The error bars in **b** and **d** represent the standard deviation of the normalized population.

pulse drives population back into $|\downarrow\rangle$. Finally, a 100 ns duration optical pulse measures the final population in $|\downarrow\rangle$. We use the ratio of the readout signal to the initialization signal as a measure of the final population in the $|\downarrow\rangle$ state, normalized to the thermal equilibrium population. By varying the frequency of the acoustic pulse and measuring the corresponding normalized population, we obtain the ODAR spectrum for the SiV spin qubit (Fig. 3c), demonstrating acoustic driving of the SiV spin. The maximum of this spectrum occurs at the transition frequency of the qubit, which is 3.43 GHz for our experiment. The ODAR spectrum has a FWHM linewidth of 48 MHz, which is broadened in part due to the frequency spread of the short 20 ns duration acoustic pulse.

**Coherent acoustic control of the SiV spin.** After matching the acoustic pulse frequency to the qubit transition frequency measured by ODAR, we vary the duration of the acoustic pulse from 0 ns to 100 ns (Fig. 4a). As the duration of the acoustic pulse increases, the normalized population in $|\downarrow\rangle$ displays Rabi oscillations (Fig. 4b), indicating coherent transfer of population between the $|\downarrow\rangle$ and $|\uparrow\rangle$ states. We fit an exponentially damped sinusoid (Supplementary Note 4) to the data to estimate the Rabi frequency. Upon varying the peak input microwave power from $-3$ dBm (0.5 mW) to 6 dBm (4 mW), we observe the expected linear increase of Rabi frequency against the square root of the power (Fig. 4b), and infer a Rabi frequency of 48 MHz with 4 mW of input microwave power. We estimate the on-chip acoustic power from the measurement of the microwave reflection S-parameter $S_{11}$ and transmission parameter $S_{21}$ (Fig. 2b). At the acoustic frequency of 3.43 GHz, we measure $|S_{11}|^2 = -0.4$ dB and $|S_{21}|^2 = -31$ dB, indicating that the peak on-chip acoustic power is between 350 $\mu$W and 3 $\mu$W, which is orders of magnitude lower than previous demonstrations of acoustic

control of defect spins[15,16]. This power is within the typical thermal load limits of dilution refrigerators, and is hence compatible with operation at millikelvin temperatures that enable longer ($\sim$10 ms) SiV spin coherence times[10].

Finally, we use this coherent acoustic control to perform Ramsey interferometry and measure the coherence time of the SiV spin qubit. We set the acoustic pulse detuning to 50 MHz from the qubit transition frequency. After estimating the duration of a $\pi/2$ acoustic pulse, we perform a Ramsey pulse sequence with two $\pi/2$ acoustic pulses separated by a varying time delay (Fig. 4c). The time constant of the exponentially decaying Ramsey fringes gives a direct measurement of the coherence time of the spin qubit, which is $T_2^* = 33 \pm 5$ ns. At our operating temperature of 5.8 K, the spin coherence time is limited by decoherence due to 50 GHz thermal phonons resonant with the orbital transition in the ground state, which follows an inverse dependence with temperature[25,26]. The measured spin coherence time agrees with previously reported results, when the temperature dependence is accounted for[17,26].

**Discussion**

In conclusion, we demonstrate coherent acoustic control of the SiV spin qubit in diamond using surface acoustic waves, and perform acoustically driven Ramsey interferometry on a single spin. The high strain susceptibility of the SiV spin qubit, arising from the ground-state orbital degeneracy[18], allows this to be performed with low acoustic power, making it compatible with operation at millikelvin temperatures which enable long ($\sim$10 ms) spin coherence times[10]. It also opens up the possibility of further improving the spin coherence time by using fast acoustic pulses for dynamical decoupling. Finally, this efficient acoustic interface could be utilized to achieve phonon-mediated

coupling of the SiV spin with a wide variety of quantum systems. In particular, placing the SiV center within a high quality factor confined mechanical mode would enable strong coupling between the spin qubit and single phonons[27]. For a diamond mechanical resonator with frequency of a few GHz and mode volume on the order of one cubed acoustic wavelength, strong coupling would be achieved at millikelvin temperatures for a quality factor of $10^3$, or at 4 K for a quality factor of $10^5$[18]. By making use of cavity optomechanical schemes[28], such a hybrid SiV-mechanical resonator system could be used to realize a high cooperativity interface between the SiV spin qubit and photons at telecommunications frequencies. Alternatively, piezoelectric schemes[29,30] could be employed to establish an interface with microwave quantum circuits such as superconducting qubits[31]. Thus our work demonstrates a promising path towards hybrid quantum systems and networks.

## Methods

**Device fabrication**. We use [100]-cut, electronic grade single-crystal diamond samples synthesized by chemical vapor deposition (CVD) from Element Six Corporation. Silicon ions ($^{28}Si^+$) are implanted on the top surface of the diamond at an energy of 150 keV and a density of $10^{10}$ cm$^{-2}$, introducing Si atoms over the entire surface at a depth of $100 \pm 18$ nm as determined by a SRIM simulation[32]. SiV centers are generated by a high-temperature (1100° C), high-vacuum annealing procedure followed by a tri-acid clean (1:1:1 sulfuric, perchloric, and nitric acids). A 1.4 μm aluminum nitride (AlN) layer is deposited on top of the diamond by RF sputtering. SAW devices are fabricated using electron beam lithography followed by evaporation of 100 nm of gold (Au) for bonding pads and 75 nm of aluminum (Al) for IDTs.

**Electrical characterization of SAW devices**. These measurements are performed at room temperature. The IDTs are contacted with RF probes, which are connected to a vector network analyzer (Agilent E8364B) for S-parameter measurements.

**Microwave impedance microscopy measurements**. The IDT is driven by a continuous-wave microwave input signal to generate surface acoustic waves. An atomic force microscopy probe scans over the surface of the device and senses the electric potential at each point[24]. The measured signal is mixed with the drive signal to coherently detect the relative amplitude and phase of the electric potential of the SAWs.

**Low temperature experiments**. Low temperature experiments are performed in a closed-cycle liquid helium cryostat (Montana Instruments Cryostation). The sample is mounted on an XYZ piezoelectric nanopositioner stack (Attocube ANPx101 and ANPz101) with a custom-made holder. The holder contains another nanopositioner with a neodymium permanent magnet (K&J Magnetics) positioned behind the sample, that can be moved relative to the sample to adjust the magnetic field. The sample is clamped on top of the holder, using Indium foil for good thermal contact. A temperature sensor in the holder estimates the sample temperature to be about 5.8 K. A pair of IDTs are wire bonded to a PCB on the holder for electrical excitation and readout of acoustic pulses. SiVs in the sample are addressed with a home-built confocal microscope that focuses light from a 520 nm laser and a tunable 737 nm laser (M-Squared SolSTiS) on the sample. The 520 nm laser is periodically switched on to reset the charge state of the SiV, while the 737 nm laser is used to resonantly excite optical transitions. The wavelength of the tunable laser is stabilized by using feedback from a wavemeter (HighFinesse WS7). Photons emitted by the SiV in the phonon side band (PSB) > 740 nm are selected by an optical long-wavelength pass filter and sent to an avalanche photo diode (APD, Excelitas) for counting.

**Pulsed measurements of SiVs**. Laser pulses at 737 nm are generated with an electro-optic intensity modulator (EOM, iXblue NIR-MX800). The EOM is biased at its half-wave voltage ($V_\pi$) and stabilized against temporal drift by using feedback with a lock-in amplifier (SRS SR830). A delay generator (SRS DG645) is used to generate the voltage pulses sent to the EOM. Acoustic pulses are produced by exciting the SAW transducers with short microwave pulses, which are generated with an arbitrary waveform generator (AWG, Tektronix AWG70001A) and amplified with an RF amplifier (Pasternack PE15A3008). The AWG clock is synchronized to the delay generator to prevent clock skew. Optical and acoustic pulses are temporally aligned by adjusting the delay of the waveform on the AWG.

## Data availability
The datasets generated and/or analysed during this study are available from the corresponding author on reasonable request.

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

## Acknowledgements

The authors thank Amirhassan Shams-Ansari, Bartholomeus Machielse, Graham Joe, Jeffrey Holzgrafe, Yeghishe Tsaturyan, and Ben Green for helpful discussions. We thank Matthew Markham and Daniel Twitchen from Element Six Ltd. for providing diamond samples. This work was supported by the Center for Integrated Quantum Materials (NSF grant No. DMR-1231319), ONR MURI on Quantum Optomechanics (Grant No. N00014-15-1-2761), NSF EFRI ACQUIRE (Grant No. 5710004174), NSF GOALI (Grant No. 1507508), Army Research Laboratory Center for Distributed Quantum Information Award No. W911NF1520067, and ARO MURI (Grant No. W911NF1810432). Device fabrication was performed at the Center for Nanoscale Systems (CNS), a member of the National Nanotechnology Coordinated Infrastructure Network (NNCI), which is supported by the National Science Foundation under NSF Award No. 1541959. CNS is part of Harvard University. The microwave impedance microscopy was supported by NSF Grant No. DMR-1707372, and was performed at University of Texas at Austin. N.S. acknowledges support by the Natural Sciences and Engineering Research Council of Canada (NSERC), the AQT Intelligent Quantum Networks and Technologies (INQNET) research program, and by the DOE/HEP QuantISED program grant, QCCFP (Quantum Communication Channels for Fundamental Physics), award number DE-SC0019219.

## Author contributions

S.Maity and L.S. designed the devices with help from Y.-I.S. S. Maity fabricated the devices with help from L.S. and C.C. S. Maity, S.B. and L.S. performed experimental measurements with help from S. Meesala and M.C. L.S., L.Z. and K.L. performed microwave impedance microscopy measurements. S. Maity, N.S. and B.P. analyzed experimental data. S. Maity and S.B. prepared the manuscript with help from all authors. M.L. supervised this project.

## Competing interests

The authors declare no competing interests.
