## [Peer Review File · Nature Communications]

REVIEWERS' COMMENTS:

Reviewer #1 (Remarks to the Author):

In this manuscript, the authors report coherent control of a single SiV spin in diamond by using surface acoustic waves. Acoustic control over spin and optical properties of solid-state defects are currently a hot topic due to the unique potentials in the field of quantum information and sensing. Diverse sets of spin defects including NV, SiV in diamond and divacancy in SiC have been studied with various types of mechanical systems such as nanomechanical resonators, bulk acoustic wave and surface acoustic wave (SAW) devices. For practical application of the acoustic control in these systems, however, it is crucial to realize strong spin-phonon coupling while maintaining long spin coherence time. In this regard, the authors argue that the hybrid SiV-SAW device in this paper can be a promising solution mainly due to the high strain susceptibility of SiV and low acoustic power of SAW at mK temperatures enabling long spin coherence times of the defect. As initial step toward the goal, they successfully demonstrated optically detected acoustic resonance, strain-controlled coherent spin population and Ramsey interferometry at 5.8 K.

This paper is well-written and of interest to the diamond and quantum information communities. Even though similar controls have been reported in recent papers [S. J. Whiteley et al., Nat. Phys. 15 490-495 (2019), H. Chen et al., arXiv:1906.06309v1], this work is the first experimental demonstration of coherent spin control based on SiV-SAW device and, therefore, is suitable to be published in Nature Communications. Below I suggest some points that need to be addressed for the publication.

1. In the conclusion paragraph, it would be better to add some discussion of how to improve the device in terms of mechanical quality factor and input coupling efficiency. For instance, currently small mechanical quality factor of $Q \sim 30$ and low input coupling efficiency of $\sim 2 - 5 \%$ can be improved with better designs of the SAW device e.g. adding grating or Bragg reflectors. Adding such outlook can be helpful for the readers to estimate realistic parameters to achieve strong coupling regime.

2. Typos

- At line 53, cenables  enables
- At line 128, Fig 3(b)  Fig. 2(b)

Reviewer #2 (Remarks to the Author):

This manuscript presents the experimental demonstration of acoustic control of electron spin states in single SiV centers in diamond. Experimental results include Rabi oscillations and Ramsey fringes using surface acoustic waves. Fast Rabi oscillations have been observed at relatively low microwave powers (~ 2 mW). The authors have also shown in the supplementary materials the use of a time domain technique to distinguish the electrical and acoustic fields.

These results are of interest and importance to both the color center and quantum information communities. Quantum spin control using surface acoustic waves opens up new avenues for fast spin control as well as device application and integration. Furthermore, the special orbital properties of the SiV ground states provide a promising and interesting platform for pursuing cavity-QED-like spin-phonon coupling. I recommend the publication of the manuscript in Nature Communications with the following comments/questions for the authors to consider:

- 1) I cannot seem to find the magnetic field strength and orientation used in the experiment.
- 2) Will the increased strain susceptibility for the spin qubit, which is needed for the acoustic spin control, correspondingly decrease the spin decoherence time (due to the increased spin-phonon

coupling for thermal phonons)?

3) In Figs. S2 and S3, the SAWs and electric fields are separated in time. I wonder if the authors have seen any effects of the oscillating electric fields on the SiV centers.

4) For the SiV center used, what is the ground-state spin-orbit splitting? I assume that the SiV center is not strongly strained.

Reviewer #1 (Remarks to the Author)

Remark *In this manuscript, the authors report coherent control of a single SiV spin in diamond by using surface acoustic waves. Acoustic control over spin and optical properties of solid-state defects are currently a hot topic due to the unique potentials in the field of quantum information and sensing. Diverse sets of spin defects including NV, SiV in diamond and divacancy in SiC have been studied with various types of mechanical systems such as nanomechanical resonators, bulk acoustic wave and surface acoustic wave (SAW) devices. For practical application of the acoustic control in these systems, however, it is crucial to realize strong spin-phonon coupling while maintaining long spin coherence time. In this regard, the authors argue that the hybrid SiV-SAW device in this paper can be a promising solution mainly due to the high strain susceptibility of SiV and low acoustic power of SAW at mK temperatures enabling long spin coherence times of the defect. As initial step toward the goal, they successfully demonstrated optically detected acoustic resonance, strain-controlled coherent spin population and Ramsey interferometry at 5.8 K.*

This paper is well-written and of interest to the diamond and quantum information communities. Even though similar controls have been reported in recent papers [S. J. Whiteley et al., Nat. Phys. 15 490-495 (2019), H. Chen et al., arXiv:1906.06309v1], this work is the first experimental demonstration of coherent spin control based on SiV-SAW device and, therefore, is suitable to be published in Nature Communications. Below I suggest some points that need to be addressed for the publication.

Response We thank the reviewer for the supportive comments and suggestions, which help us improve this manuscript. We respond to each of Reveiwer #1's points below.

Remark 1. *In the conclusion paragraph, it would be better to add some discussion of how to improve the device in terms of mechanical quality factor and input coupling efficiency. For instance, currently small mechanical quality factor of $Q \sim 30$ and low input coupling efficiency of $\sim 2 - 5\%$ can be improved with better designs of the SAW device e.g. adding grating or Bragg reflectors. Adding such outlook can be helpful for the readers to estimate realistic parameters to achieve strong coupling regime.*

Response This device is not designed to be a resonator. For SAW resonators with Bragg gratings, quality factors of 10^5 have been demonstrated at millikelvin temperatures [R. Manenti et al., Phys. Rev. B **93**, 041411 (2016)]. However, achieving strong spin-phonon coupling for the SiV will require resonators with small mode volumes in addition to high quality factors, as discussed in [S. Meesala et al., Phys. Rev. B **97**, 205444 (2018)]. We have added a sentence with some of the estimates from this reference to our conclusion paragraph.

Remark 2. *Typos*

- At line 53, *cenables* \rightarrow *enables*
- At line 128, *Fig 3(b)* \rightarrow *Fig. 2(b)*

Response We thank the reviewer for pointing out these mistakes. They have been fixed.

Reviewer #2 (Remarks to the Author)

Remark *This manuscript presents the experimental demonstration of acoustic control of electron spin states in single SiV centers in diamond. Experimental results include Rabi oscillations and Ramsey fringes using surface acoustic waves. Fast Rabi oscillations have been observed at relatively low microwave powers (~ 2 mW). The authors have also shown in the supplementary materials the use of a time domain technique to distinguish the electrical and acoustic fields.*

These results are of interest and importance to both the color center and quantum information communities. Quantum spin control using surface acoustic waves opens up new avenues for fast spin control as well as device application and integration. Furthermore, the special orbital properties of the SiV ground states provide a promising and interesting platform for pursuing cavity-QED-like spin-phonon coupling. I recommend the publication of the manuscript in Nature Communications with the following comments/questions for the authors to consider:

Response We thank the reviewer for the supportive comments and suggestions, which help us improve this manuscript. We respond to each of Reveiwer #2's comments below.

Remark 1) *I cannot seem to find the magnetic field strength and orientation used in the experiment.*

Response The magnetic field is oriented approximately along the z -direction normal to the diamond surface ([001] crystal direction). The field strength is estimated to be about 0.19 T by fitting a theoretical model to the positions of the C1-C4 lines.

Remark 2) *Will the increased strain susceptibility for the spin qubit, which is needed for the acoustic spin control, correspondingly decrease the spin decoherence time (due to the increased spin-phonon coupling for thermal phonons)?*

Response At the temperature (5.8 K) at which this experiment is performed, the contribution of 50 GHz thermal phonons to spin decoherence is much higher than the contribution of few GHz thermal phonons resonant with the qubit transition. This allows us to increase the qubit strain susceptibility with an off-axis magnetic field, without adversely affecting the spin coherence via direct qubit-phonon relaxation.

The eventual outlook is to operate at mK temperatures [D. D. Sukachev et al., Phys. Rev. Lett. **119**, 223602 (2017)] where acoustic modes at the frequencies of interest have near-zero thermal occupation. In this regime, qubit decoherence will be determined by spontaneous emission into the acoustic density of states resonant with the qubit transition frequency, so the reviewer's statement is correct, but only in bulk. For an SiV in a GHz frequency mechanical resonator with a phononic bandgap [M. J. Burek et al., Optica **3**, 1404-1411 (2016)], at mK temperatures the qubit will couple dominantly to one acoustic mode that is well-isolated from the bulk. In this way, one can achieve the ideal combination of qubit coherence and large acoustic coupling strength.

Remark 3) *In Figs. S2 and S3, the SAWs and electric fields are separated in time. I wonder if the authors have seen any effects of the oscillating electric fields on the SiV centers.*

Response No, we have not. In Supplementary Fig. 2, the electrical response precedes the SAW response by about 21 ns. If there was any significant effect of the oscillating electric field on the SiV, it would occur about half of this value (10 ns) before the effect of the acoustic pulse, as the SiV is located halfway between the two transducers. However, in Supplementary Fig. 4, we do not observe a significant change in photon counts 10 ns before the response to the acoustic pulse. Additionally, in Supplementary Fig. 5, we do not observe a significant change in photon counts 21 ns before the response to the acoustic pulse from the left transducer. Therefore we infer that the oscillating electric field does not have a significant effect.

Remark 4) *For the SiV center used, what is the ground-state spin-orbit splitting? I assume that the SiV center is not strongly strained.*

Response No, the SiV is not strongly strained. Although we did not determine the position of the D optical transition line for this center, we estimate a ground-state splitting of about 51 GHz by fitting a theoretical model to the positions of the C1-C4 lines.